# Anger Suppression and Rumination Sequentially Mediates the Effect of Emotional Labor in Korean Nurses

**DOI:** 10.3390/ijerph16050799

**Published:** 2019-03-05

**Authors:** Ji Eun Kim, Jeong Hoon Park, Soo Hyun Park

**Affiliations:** Department of Psychology, Yonsei University, Seoul 03722, Korea; alalvh1004@hotmail.com (J.E.K.); jpark5664@naver.com (J.H.P.)

**Keywords:** nurses, emotional labor, anger, depression, anxiety

## Abstract

The degree of emotional labor in nurses has been associated with negative physical and psychological health indices. The purpose of this study was to examine the relationship between emotional labor and depressive and anxiety symptoms in nurses. Specifically, the study addressed the question of whether anger suppression and anger rumination sequentially mediated the relationship. A total of 99 nurses was recruited from a university hospital in Korea. The questionnaires included instruments assessing emotional labor, anger suppression, anger rumination, as well as depressive and anxiety symptoms. Consistent with our hypothesis, there was a significant indirect effect of anger suppression and anger rumination on the relationship between emotional labor and depressive symptoms, as well as on the relationship between emotional labor and anxiety symptoms. The nurses’ degree of emotional labor, anger suppression, and anger rumination was associated with psychological adjustment. Thus, the impact of nurses’ negative affect needs to be adequately addressed, as inadequate resolution of anger may increase their vulnerability to experience depressive and anxiety symptoms. These findings may contribute to developing a strategy for enhancing nurses’ psychological health.

## 1. Introduction

In recent years, the psychological adjustment of the emotional laborers has received a great deal of attention. The need for research on emotional labor experienced by nurses has become inevitable. Emotional labor demands the induction or suppression of feelings to maintain a particular outward appearance [1]. Also, the service provision and emotional expression of nurses needs to appear genuine for patients under circumstances where extreme or negative feelings have been triggered. Therefore, nursing involves emotional caring that requires a balance of emotional engagement and detachment, as hospitals are increasingly demanding emotional labor [2]. Nurses are likely to encounter difficulties in regulating their emotions due to emotional dissonance—a discrepancy between emotions they are actually experiencing and emotions they are required to express.

Korean nurses may struggle with difficulties in emotional labor due to the suppression of emotions in order to conform to patients’ and organizational demands. Research suggests that Korean nurses report a greater degree of anger and stress than other professions. Among nurses who reported significant stress, 44.4% chose anger as the most frequently experienced emotion, and they suppressed anger more than they expressed anger [3]. It was concluded that nurses tend to experience negative emotions, in particular anger, and that the emotions are usually suppressed due to job demands.

In particular, anger suppression significantly affects psychological adjustment. Previous studies suggested that individuals who engage in greater anger suppression usually display a higher degree of depression and anxiety [4]. Cox, Van Velsor, and Hulgus also reported that women who cannot adequately express anger and opt to divert anger via avoidance of anger awareness or by using indirect means to cope with anger are more vulnerable to depression and anxiety than women who express anger [5]. From these findings, it can be assumed that nurses, who tend to experience frequent anger but also exhibit a tendency to suppress anger, may be particularly vulnerable to depressive and anxiety symptoms.

The suppression of negative emotions is associated with rumination, which may eventually cause, maintain, or increase depressive and anxiety symptoms [6]. The potential role of rumination in the context of emotional suppression is partially supported by research examining the relationship between rumination and thought suppression. These studies showed that consciously suppressing unwanted thoughts can ironically increase the availability of the suppressed thoughts [7]. Thus, it is possible that suppression may fuel rumination, which may subsequently increase susceptibility to experiencing depressive and anxiety symptoms.

Rumination is not only involved with a low activity mood, such as sadness, but it may also be involved with hyperactive mood states, such as anger. Rumination in this context has been termed “anger rumination”. Anger rumination can be described as the tendency to persistently deliberate on past events that had triggered anger [8]. Although similar in concept, anger rumination has been suggested as an independent cognitive process which follows anger provocation, thereby maintaining or enhancing anger mood and anger-related behavior problems. In addition, higher use of rumination as a coping strategy in an anger-provoking situation contributes to depression and anxiety [8].

In summary, extant research indicates that in the process of emotional labor exerting influence on the degree of depressive and anxiety symptoms, the anger-related mechanism may play a significant role. Based on the assumption that there is an integrative causal association between emotional labor, anger suppression and rumination, as well as depressive and anxiety symptoms, the present study sought to empirically examine this possibility. In other words, nurses are required to suppress their anger evoked from emotional labor, and ruminating about anger-provoking incidents and anger mood may likely intensify the problem and possibly contribute to depressive and anxiety symptoms. The present study specifically addressed the question of whether anger suppression and anger rumination sequentially mediate the relationship in order to examine its causal relationship.

## 2. Materials and Methods

### 2.1. Study Design and Participants

The current study adopted a cross-sectional self-report survey design in order to examine the sequential mediating effect of anger suppression and anger rumination in the relationship between emotional labor and depressive and anxiety symptoms of nurses. The study protocol was approved by the Yonsei University Institutional Review Board (Approval No. 7001988-201612-SB-102-02).

The participants in the study were 99 nurses who worked for at least six months at a university hospital in the Republic of Korea. They were recruited during January of 2017. Data were collected via the distribution of self-report questionnaires given to all nurses in the hospital who expressed interest in participating in the study. Of the 110 surveys distributed, 99 surveys were returned. All 99 surveys were usable for further data analysis without problems concerning missing data or multiple responses. Of the 99 respondents, the mean age was 30 years (standard deviation (SD) = 7.17) and 98.0% were women. The majority of participants were staff nurses (*n* = 90, 90.9%) who had been working in their current position between 12 and 60 months (*n* = 51, 51.5%). Their work pattern consisted mostly of triple shifts (*n* = 89, 89.9%), whereas nine participants (9.1%) reported a non-shifting work schedule (Table 1). The work shifts consisted of a day shift (7:30–15:00), an evening shift (15:00–22:00), and a night shift (22:00–7:30).

### 2.2. Measures

Emotional Labor—The Korean-Emotional Labor Scale (K-ELS), developed and validated by Jang et al., and based on the characteristics of emotional labor in Korea, was used to measure emotional labor [9]. This tool was composed of 26 items with five subscales, and the total score was used in the present study. Sample items of the K-ELS are “I make efforts not to express negative feelings towards customers” and “I get hurt in the process of facing customers.” The items were rated on a four-point Likert scale ranging from 1 (strongly disagree) to 4 (strongly agree). In the present study, the Cronbach’s alpha coefficient for the K-ELS scale was 0.87.

Anger Suppression—The Korean State-Trait Anger Expression Inventory (STAXI-K) [10], which is a modified and validated version of the State-Trait Anger Expression Inventory (STAXI) [11], was used to measure anger suppression. This tool is composed of 10 items for each trait anger and state anger, and eight items for each anger-in, anger-out, and anger-control. The items are rated on a four-point Likert scale ranging from 1 (strongly disagree) to 4 (strongly agree). For this study’s purpose, only the anger-in subscale from the state anger scale was used. Sample state anger scale items of the STAXI-K are “I am angry” and “I yell.” A higher anger-in total score indicates more frequent anger suppression. In the present study, the Cronbach’s alpha coefficient for the anger-in subscale was 0.79.

Anger Rumination—The Korean Anger Rumination Scale (K-ARS) [12], which is a modified and validated version of the Anger Rumination Scale (ARS) [13], was used to measure anger rumination. The items are rated on a four-point Likert scale ranging from 1 (almost never) to 4 (almost always in terms of how well the items resemble their beliefs about themselves; higher scores correspond to a greater degree of anger rumination. In the present study, the Cronbach’s alpha coefficient for the K-ARS scale was 0.95.

Depressive Symptoms—Depressive symptoms were assessed using the Korean version of the Center for Epidemiologic Studies Depression Scale (CES-D) [14], originally developed by the National Institute of Mental Health [15]. This is a 20-item self-report questionnaire that measures the frequency and severity of depressive symptoms experienced within the last week. The items are rated on a four-point Likert scale ranging from 0 (very rare, i.e., less than one day during one week) to 3 (most of the time, i.e., more than five days during one week). In the present study, the Cronbach’s alpha coefficient was 0.93.

Anxiety Symptoms—The State-Trait Anxiety Inventory (STAI), originally developed by Spielberger, Gorsuch, Lushene, Vagg, and Jacobs [16] and adapted for use in Korea [17], was used to measure anxiety symptoms. It measures state anxiety (20 items) and trait anxiety (20 items), where state anxiety represents anxiety that is temporary and changing, and trait anxiety represents anxiety that is permanent and stable. For the purpose of this study, only 20 items measuring trait anxiety were used. The items are rated on a four-point Likert scale ranging from 1 (not at all) to 4 (very much). In the present study, the Cronbach’s alpha coefficient for trait anxiety was 0.91.

### 2.3. Data Analysis

The sequential multiple mediation analysis suggested by Hayes was conducted using PROCESS Macro for SPSS (IBM Corp., Armonk, NY, USA) [18]. The significance of all the possible indirect effects from the study’s model were examined using the bootstrapping method of 5000 resamples. The 95% confidence intervals that did not include zero indicated a significant indirect effect. Previous research suggested that work shift pattern, length of work experience, and job position may influence nurses’ degree of depressive symptoms [19,20]. Therefore, these demographic variables were controlled as covariates in the statistical analyses.

## 3. Results

### 3.1. Descriptive Statistics and Intercorrelations between Variables

Table 2 presents descriptive statistics and intercorrelations between the study variables. Emotional labor was positively correlated with both anger suppression (*r* = 0.43, *p* < 0.001) and anger rumination (*r* = 0.30, *p* = 0.003). Anger suppression demonstrated significant positive correlations with depressive (*r* = 0.30, *p* = 0.003) and anxiety (*r* = 0.24, *p* = 0.020) symptoms. Anger rumination was also positively associated with depressive (*r* = 0.42, *p* < 0.001) and anxiety (*r* = 0.35, *p* = 0.001) symptoms. However, the degree of emotional labor did not show significant correlations with either depressive (*r* = 0.13, *p* = 0.215) or anxiety (*r* = 0.11, *p* = 0.289) symptoms.

### 3.2. Sequential Mediation Analysis

Sequential mediation analyses were conducted after controlling for the work shift pattern, the length of work experience, and the job position as aforementioned (Table 3).

Depression—There was a significant total indirect effect, which was the sum of all possible indirect effects in the model between emotional labor and depressive symptoms (effect = 0.16; confidence interval (CI): from 0.05 to 0.33), but a non-significant direct effect (effect = −0.07, *t* = −0.63, *p* = 0.531). Regarding each indirect effect, there was a significant indirect effect indicating that the path from anger suppression to anger rumination sequentially mediated the relationship between emotional labor and depressive symptoms (X→M1→M2→Y: effect = 0.12; CI: from 0.04 to 0.26). The indirect effects from emotional labor to anger suppression to depressive symptoms (X→M1→Y: effect = 0.01; CI: from −0.12 to 0.14) and from emotional labor to anger rumination to depressive symptoms (X→M2→Y: effect = 0.04; CI: from −0.03 to0.18) were not significant. The coefficients and standard error (SE) for each path are presented in Figure 1.

Anxiety—The total indirect effect was significant (effect = 0.12; CI: from 0.02 to 0.27), but there was a non-significant direct effect (b = −0.03, t = −0.27, *p* = 0.790). Regarding each indirect effect, there was a significant indirect effect of emotional labor on anxiety through anger suppression and anger rumination, again indicating sequential mediation (X→M1→M2→Y: effect = 0.09; CI: from 0.02 to 0.22). The indirect effects from emotional labor to anger suppression to anxiety (X→M1→Y: effect = −0.02; CI: from −0.14 to 0.12) and the indirect effect from emotional labor to anger rumination to anxiety (X→M2→Y: effect = 0.04; CI: from −0.02 to 0.16) were not significant. The coefficients and SE for each path are presented in Figure 2.

## 4. Discussion

Consistent with our hypothesis, the results indicated that emotional labor had an indirect sequential effect on depressive and anxiety symptoms through anger suppression and anger rumination, and the direct effect of emotional labor on depression and anxiety was not significant after controlling for anger suppression and anger rumination. Hence, the nurses’ emotional labor itself may not serve as a risk factor per se for psychological maladjustment. In addition, anger suppression or rumination may not lead to emotional problems. The results of the present study indicate that nurses who experience a high degree of emotional labor, who tended to suppress anger and who also ruminated about the causes and effects of anger-provoking situations, may maintain or intensify anger mood, which ultimately increased their vulnerability to experience depressive and anxiety symptoms.

Interestingly, the indirect effect of anger suppression alone was not significant. Specifically, emotional labor was associated with anger suppression, but the pathway from anger suppression to depressive and anxiety symptoms was not significant. It is still an ongoing discussion as to whether emotional suppression always demonstrates a negative effect on psychological adjustment. There are results from meta-analytic studies which suggest that emotional suppression exhibits a negative impact on an individual’s psychological health, but at the same time, there have also been studies that have reported on adaptive aspects of emotional suppression [21,22]. Thus, it cannot be assumed that simply suppressing negative emotions always leads to psychological maladjustment. Moreover, the indirect effect of anger rumination on the path from emotional labor to depressive and anxiety symptoms was not significant. Specifically, anger rumination had a significant effect on depressive and anxiety symptoms, but the path from emotional labor to anger rumination was not significant. Based on our results, the cognitive process of repetitively recalling anger-provoking situations (rumination) may have a significant impact in aggravating depressive and anxiety symptoms, especially when anger is suppressed.

In spite of the results from previous studies which reported that emotional labor has a negative effect on psychological adjustment [23], emotional labor and depressive and anxiety symptoms were not significantly correlated in the current study. This result may be related to the characteristic of emotional labor. The two main emotion regulation strategies used by emotional laborers are surface acting, (i.e., displaying nongenuine desired emotions) and deep acting, (i.e., modifying their genuine emotions to actually feel the desired and situationally required emotions) [1]. According to Drach-Zahavy, Yagil, and Cohen, surface acting requires constant effort to suppress negative emotion, which leads to the depletion of psychological resources and as a result impedes one’s well-being [24]. On the other hand, deep acting may work as a protective factor in psychological adjustment, since it also involves efforts to maintain a positive view in negative or stressful situations. Empirical research has also demonstrated that this kind of cognitive effort may have a positive effect. In addition, in social interactions, deep acting may assist in the formation of relatively positive relationships and increase the opportunity to receive positive feedback from others by presenting authentic emotions during interactions [25]. If this effect is applied to nurses, deep acting may contribute to the formation of relatively positive interactions with patients, and nurses may also experience job satisfaction through potential positive feedback. These opposite possibilities may simultaneously affect depressive and anxiety symptoms.

The present study may provide valuable information for constructing a specific intervention strategy to protect nurses’ psychological adjustment. More specifically, training in the use of different kinds of adaptive emotional regulation strategies other than anger suppression and anger rumination may be needed. For example, cognitive reappraisal was found to be more effective at reducing and handling anger than efforts to suppress anger [26]. Furthermore, positive reappraisal (i.e., searching for positive aspects or meaning instead of ruminating about anger-provoking events) and positive refocusing (i.e., focusing on pleasant and positive experiences unrelated to the experienced negative events) may be effectively utilized [27]. Supplementing clinically validated effective interventions for depressive and anxiety disorders, such as cognitive behavior therapy (CBT), with such emotion regulation training components may further effectively reduce nurses’ psychological discomfort such as anger and depressive and anxiety symptoms.

## 5. Limitations and Future Directions

As reported in the meta-analysis results of Hülsheger and Schewe, there may be a significant moderating effect of cultural differences in the relationship between emotional labor and psychological distress [28]. Eastern cultures are characterized as an interdependent culture that values conformity, and the suppression of emotional expression for the sake of group harmony is frequently demanded [29]. Thus, there may exist a mediating effect of cultural differences on the sequential mediating effect of anger suppression and anger rumination found in the present study. Hence, the results of the current study need to be replicated through cross-cultural research. If differences are found, research identifying the cultural factors that contribute to the differences also needs to be conducted.

There is a possibility that the university hospital worked as a protective factor compared to other work environments. According to previous studies, work organization, structure, and policy have a significant impact on employees’ anger [30]. University hospitals have relatively well-established organizational support and a protective system that may work in some degree as a protective factor for the nurses’ anger [10]. Thus, examining the organizational characteristics of nurses working at private hospitals and other emotional laborers that may influence the result is suggested for further research.

Also, since the current study was conducted using cross-sectional data, we cannot confirm long-term variation trends in anger suppression and rumination and subsequent depression and anxiety. It is uncertain whether it is a temporary or a long-term effect that anger suppression and rumination have on depressive and anxiety symptoms. A study by Takebe et al. longitudinally examined the relationship between anger suppression and anger rumination [9]. Thus, future studies that examine the long-term effects on depressive and anxiety symptoms are suggested, based on our finding that these variables are significantly associated.

## 6. Conclusions

The present study suggested the existence of a potential mechanism by which anger that is evoked from emotional labor may be related to depressive and anxiety symptoms. Thus, hospitals should acknowledge that it is critical to manage nurses’ anger through the construction of a specific intervention strategy to protect nurses’ psychological adjustment.

## Figures and Tables

**Figure 1 ijerph-16-00799-f001:**
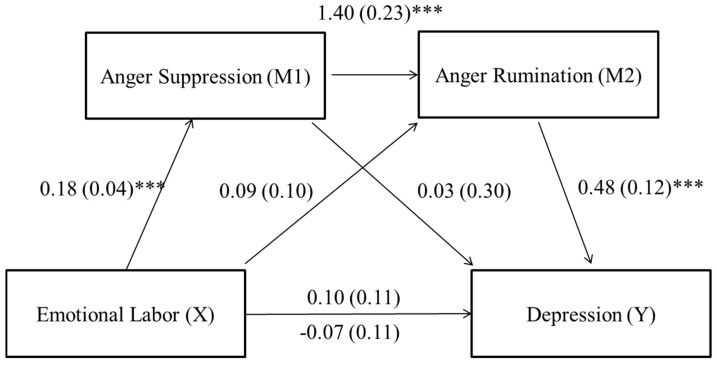
Sequential mediation model for emotional labor and depression via anger suppression and anger rumination. Unstandardized path coefficients and SE indicated above. The coefficient appearing above the line connecting emotional labor and depression represents the total effect and the coefficient below the line represents the direct effect. *** *p* < 0.001.

**Figure 2 ijerph-16-00799-f002:**
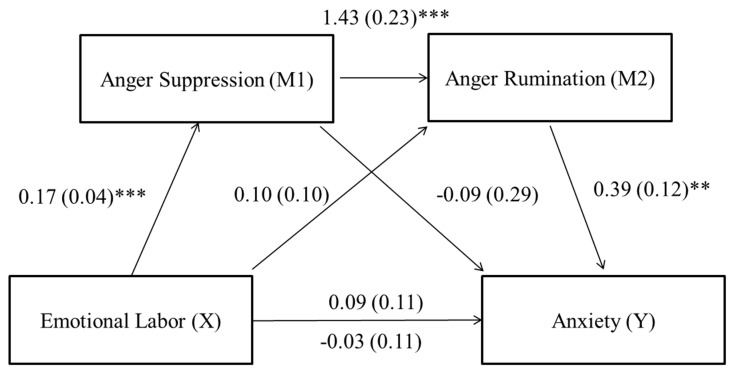
Sequential mediation model for emotional labor and anxiety via anger suppression and anger rumination. The unstandardized path coefficients and SE are indicated above. The coefficient appearing above the line connecting emotional labor and anxiety represents the total effect and the coefficient below the line represents the direct effect. ** *p* < 0.01, *** *p* < 0.001.

**Table 1 ijerph-16-00799-t001:** Summary of demographic characteristics (*N* = 99).

Characteristics	*n* (%)
Age (years)	
≤29	65 (65.7)
30–39	23 (23.3)
≥40	11 (11.0)
Education level	
Associate degree	21 (21.2)
Bachelor degree	63 (63.7)
Attending graduate school	4 (4.0)
Master or doctorate degree	10 (10.1)
Other	1 (1.0)
Position department	
General ward	72 (72.7)
Intensive care unit	10 (10.1)
Other	17 (17.2)
Length of work experience in current position (months)	
<12	28 (28.3)
12–60	51 (51.5)
61–120	15 (15.1)
≥121	6 (6.1)
Work schedule	
Non-shift/day shift	9 (9.1)
Double shift	1 (1.0)
Triple shift	89 (89.9)
Current position	
Staff nurse	90 (90.9)
≥Charge nurse	9 (9.1)

**Table 2 ijerph-16-00799-t002:** Summary of intercorrelations, means, and standard deviations for study variables.

Measure	1	2	3	4	5	Mean	*SD*
	*r* (*p*)						
1. AngerSup	1					14.21	3.90
2. KARS	0.59 (<0.001)	1				29.99	9.26
3. CES-D	0.30 (0.003)	0.42 (<0.001)	1			18.67	9.81
4. STAI	0.24 (0.020)	0.35 (0.001)	0.84 (<0.001)	1		47.48	9.29
5. K-ELS	0.43 (<0.001)	0.30 (0.003)	0.13 (0.215)	0.11 (0.289)	1	69.31	9.07

Note: Anger suppression (AngerSup); Korean version of the Center for Epidemiologic Studies Depression Scale (CES-D); Korean Anger Rumination Scale (KARS); Korean-Emotional Labor Scale (K-ELS); Korean State-Trait Anxiety Inventory (STAI).

**Table 3 ijerph-16-00799-t003:** Sequential mediation effect of anger suppression and anger rumination in the relationship between emotional labor and depression/anxiety.

Dependent Variables		Effect	SE	*t*	*p*	LLCl	ULCl
Depression (Y)	Total effect	0.10	0.11	0.85	0.396	−0.13	0.32
	Total indirect effect	0.16	0.07			0.05	0.33
	Indirect paths						
	X→M1→Y	0.01	0.07			−0.12	0.14
	X→M2→Y	0.04	0.05			−0.03	0.18
	X→M1→M2→Y	0.12	0.06			0.04	0.26
	Direct effect	−0.07	0.11	−0.63	0.531	−0.29	0.15
Anxiety (Y)	Total effect	0.09	0.11	0.86	0.790	−0.12	0.30
	Total indirect effect	0.12	0.06			0.02	0.27
	Indirect paths						
	X→M1→Y	−0.02	0.06			−0.14	0.12
	X→M2→Y	0.04	0.04			−0.02	0.16
	X→M1→M2→Y	0.09	0.05			0.02	0.22
	Direct effect	−0.03	0.11	−0.27	0.790	−0.24	0.19

Note: Emotional labor (X); anger suppression (M1); anger rumination (M2); lower levels for 95% confidence interval (LLCI); upper levels for 95% confidence interval (ULCI).

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
