# Peer review of "Anger Suppression and Rumination Sequentially Mediates the Effect of Emotional Labor in Korean Nurses"

_ijerph, 2019, doi:10.3390/ijerph16050799_

Round 1

Reviewer 1 Report

Anger suppression and rumination sequentially mediates the effect of emotional labor in Korean nurses

The subject itself is interesting but it has been intensively studied. So there is a need to have new perspective on the subject.

Introduction

I would like to see introduction as a part of manuscript where authors tell from their point of view why this current study is needed. I got a feeling that this introduction is very much only showing to the reader what has been done. The personal insight is lacking. It is not recommended to cite nearly the sentence from the book by Hochschild. I like mentioning eastern culture, but this interesting idea is not used effectively later in the manuscript and only study by Hülsheger and Schewe is mentioned in discussion. I think this issue potentially has more impact.

It is stated that the current study addresses the question of whether anger suppression and anger rumination sequentially mediate the relationship in order to examine its causal relationship. I don’t totally understand the purpose because authors state in the discussion part that “since the current study was conducted using cross-sectional data, we cannot confirm the causal relationship between the variables”.

Materials and methods

Authors total skip the question how the nurses were selected? Did all the nurses who received the questionnaire answer? Did all the nurses answer to all the questions? Why the authors used nurses who have worked so little time? Does this cause some impact for the results? Authors use later work shif pattern, length of work experience and job position. Why these are not shown? Authors don’t tell in what department the nurses worked.  I would like to see these very fundamental questions when describing the participants.

Reference 11 was only available by Korean language. It should be substituted by English reference or omitted from the methodology.

Discussion

I tried to think for the clinical point of view what the following sentence means:”The results that indicate emotional labor has an indirect effect on depressive and anxiety symptoms through anger suppression and anger, without a significant effect of emotional labor has on depression and anxiety.

Authors continue that the present study may provide valuable information for constructing a specific intervention strategy to protect nurses’ psychological adjustment. Authors continue then to adaptive emotional regulation strategies but it remains unclear for the reader what is the “specific intervention strategy especially for nurses”.

Conclusion

Authors last sentence is the following: “Future research should investigate specific elements that affect nurses’ psychological health and provide ways to decrease their psychological discomfort.” This sentence is not telling me what the authors suggest to study next.

Author Response

Comments from Reviewer #1

Revisions

1. The subject itself is interesting but it   has been intensively studied. So there is a need to have new perspective on   the subject.

First,   the authors would like to thank the reviewer for a comprehensive review of   our manuscript. We believe that Reviewer #1’s comment is quite valid. To this   end, we have revised the manuscript to highlight what at least the authors   believe to be a different perspective from extant research related to emotional   labor. The specific revised content is described below. Thank you.

2.   I would like to see introduction as a part of   manuscript where authors tell from their point of view why this current study   is needed. I got a feeling that this introduction is very much only showing   to the reader what has been done. The personal insight is lacking. It is not   recommended to cite nearly the sentence from the book by

In   response to Reviewer #1’s suggestion, we have revised our manuscript   introduction section (pages 1-2).

First,   the authors deleted reference to Hoschild and review of emotional labor.   Based on theoretical background, the authors added discussion regarding the   process by which our hypotheses were generated. We have also deleted   unnecessary sentences or added new discussion for the sake of logical   progression.

For   example, we added on page 2 lines 64-68 the following sentences:

In summary, extant   research indicates that in the process of emotional labor exerting influence   on degree of depressive and anxiety symptoms, anger-related mechanism may   play a significant role. Thus, the present study assumed an integrative   causal association between emotional labor, anger suppression and rumination,   and depressive and anxiety symptoms and sought to empirically examine this   possibility. In other words,…”

3.  I like   mentioning eastern culture, but this interesting idea is not used effectively   later in the manuscript and only study by Hülsheger and Schewe is mentioned   in discussion. I think this issue potentially has more impact.

We thank   Reviewer #1 for such an encouraging feedback. To address this question, the   authors have added discussion regarding the potential role of the eastern   culture on page 8, lines 236 to 243 with additional referencing. We sought to   emphasize the potential need to examine culture in future research. The   specific addition is as follows:

“Eastern cultures are characterized as   an interdependent culture that values conformity and suppression of emotional   expression for the sake of group harmony is frequently demanded   [29]. Thus, there may exist a mediating effect of cultural differences on the   sequential mediating effect of anger suppression and anger rumination found   in the present study.”

4.  It is stated   that the current study addresses the question of whether anger suppression   and anger rumination sequentially mediate the relationship in order to   examine its causal relationship. I don’t totally understand the purpose   because authors state in the discussion part that “since the current study   was conducted using cross-sectional data, we cannot confirm the causal   relationship between the variables”.

After reading   Reviewer #1’s comment, the authors see how our previous writing may have   caused confusion. Our discussion regarding causal relationship has been   changed as follows (page 8, line 251):

“Also, since the current study was   conducted using cross-sectional data, we cannot confirm long-term variation   trends in anger suppression and rumination and subsequent depression and   anxiety.”

5.    Authors total   skip the question how the nurses were selected? Did all the nurses who   received the questionnaire answer? Did all the nurses answer to all the   questions? Why the authors used nurses who have worked so little time? Does this   cause some impact for the results? Authors use later work shift pattern,   length of work experience and job position. Why these are not shown? Authors   don’t tell in what department the nurses worked.  I would like to see these very fundamental   questions when describing the participants.

We   thank Reviewer #1 for such such a careful review. In response to the comment,   we have added information regarding the participant recruitment and selection   process. We hav also added demographic information regarding the participants   to help the reviewers understand the sample better (e.g, work department, work   shift information, length of work experience) (page 2, line 64 to page 3,   line 91; Table 1).

“Participants were 99 nurses working   for at least 6 months at a university hospital in the Republic of Korea who   were recruited during January of 2017. Data were collected via distribution   of self-report questionnaires to all nurses in the hospital who expressed   interest in participating in the study. Of the 110 surveys distributed, a   total of 99 surveys were returned. All 99 surveys were usable for further   data analysis without problems concerning missing data nor multiple   responses. Of the 99 respondents, the mean age was 30 years (SD=7.17) and   98.0% were women. Majority of participants were staff nurses (n=90, 90.9%)   who had been working in the current position between 12 to 60 months (n=51,   51.5%). Their work pattern consisted mostly of triple shifts (n=89, 89.9%),   whereas nine participants (9.1%) reported a non-shifting work schedule (Table   1). The work shifts consisted of day shift (7:30-15:00 hours), evening shift   (15:00-22:00 hours), and night shift (22:00-7:30 hours).”

Table   1.   Summary of Demographic Characteristics (N=99).

Characteristics  n (%)Age ≤2965 (65.7)30-3923 (23.3)≥4011 (11.0)Education     level Associate     degree21 (21.2)Bachelor     degree63 (63.7)Attending     graduate school4 (4.0)Master or     doctoral degree10 (10.1)Other1 (1.0)Position     department General ward72 (72.7)Intensive care     unit10 (10.1)Other17 (17.2)Length     of work experience in current position (months) <1228 (28.3)12-6051 (51.5)61-12015 (15.1)≧1216 (6.1)Work     schedule Non-shift/day     shift9 (9.1)Double shift1 (1.0)Triple shift89 (89.9)Current     position Staff nurse90 (90.9)≧Charge nurse9 (9.1)

6.    Reference 11 was   only available by Korean language. It should be substituted by English   reference or omitted from the methodology.

The   particular reference (Reference #11) mentioned by Reviewer #1 is   unfortunately only available in Korean language. The authors are not able to   substitute this particular reference because the reference concerns the   development and validation of the Korean-Emotional Labor Scale (K-ELS) that was   used in our research. This particular scale was developed for the sole   purpose of examining and measuring emotional labor in the Korean culture.   This was also the reason we chose this particular instrument. In other words,   the authors believed that this was the only instrument developed with the   goal of measuring emotional labor in the Korean cultural and work context. We   ask for the reviewer’s understanding. Thank you.

7.   I tried to think for the clinical point of view what the following sentence   means: “The results that indicate emotional labor has an indirect effect on   depressive and anxiety symptoms through anger suppression and anger, without   a significant effect of emotional labor has on depression and anxiety.

We   have revised the particular sentence for grammatical correctness and English   clarity. The sentence has been revised as follows to more clearly underline   the clinical implication (page 6, line 185):

“Consistent   with our hypothesis, the results indicated that emotional labor has an   indirect sequential effect on depressive and anxiety symptoms through anger   suppression and anger rumination and the direct effect of emotional labor on depression and anxiety   was not significant after controlling for anger suppression and anger   rumination.”

8.    Authors continue that the   present study may provide valuable information for constructing a specific   intervention strategy to protect nurses’ psychological adjustment. Authors   continue then to adaptive emotional regulation strategies but it remains   unclear for the reader what is the “specific intervention strategy especially   for nurses”.

We   thank the reviewer for such helpful suggestions. We have provided examples   related to “specific intervention strategy” on page 7, line 230 as follows:

“Supplementing clinically validated   effective interventions for depressive and anxiety disorders such as   cognitive behavior therapy (CBT) with such emotion regulation training   components may further effectively reduce nurses’ psychological discomfort   such as anger, depressive and anxiety symptoms.”

9.    Authors last sentence is   the following: “Future research should investigate specific elements that   affect nurses’ psychological health and provide ways to decrease their   psychological discomfort.” This sentence is not telling me what the authors   suggest to study next.

After   careful consideration following the reviewer’s comment, the authors concluded   that ending the Conclusion section with such a sentence may be inappropriate.   As such, we have deleted the particular sentence (page 8, line 258). Thank   you.

Reviewer 2 Report

I would like to congratulate you on a nice paper!  It is efficiently written and gets right to the point.  Nicely done.  

I do have a few recommendations I believe will strengthen the paper:

Proofreading for language/grammar errors.  Not major work to be done here, but there are some spots that are difficult to follow for a primary English speaker

I would like to see some more information on your methods.  Specifically, how was your sample derived?  How were your instruments administered/results collected?

You mention that your paper should provide guidance for "developing a strategy for enhancing nurses' psychological health:.  I would like to here more discussion of what you envision in this area.  

I think a separate section outlining the limitations of the study and  providing more discussion on your call for future directions of research would be helpful.  Not a big issue, but I think a bit more discussion here would help.

Author Response

Comments from Reviewer #2

Revisions

1.    I would like to   congratulate you on a nice paper!  It   is efficiently written and gets right to the point.  Nicely done. 

I   do have a few recommendations I believe will strengthen the paper:

The authors thank Reviewer #2 for such kind   and encouraging comments. We believe that revising our manuscript based on   the reviewers’ comments will improve the quality of our manuscript   significantly. We thank Reviewer #2. We have revised our manuscript based on   your comments as follows.

2.    Proofreading for   language/grammar errors.  Not major   work to be done here, but there are some spots that are difficult to follow   for a primary English speaker

We have reviewed our manuscript again to   correct any grammatical error and problems related to clarity and flow of   logical progression. Thank you.

3.    I would like to   see some more information on your methods.    Specifically, how was your sample derived?  How were your instruments   administered/results collected?

We   thank Reviewer #2 for such such a careful review. In response to the comment,   we have added information regarding the participant recruitment and selection   process. We hav also added demographic information regarding the participants   to help the reviewers understand the sample better (e.g, work department, work   shift information, length of work experience) (page 2, line 64 to page 3,   line 91; Table 1).

“Participants were 99 nurses working   for at least 6 months at a university hospital in the Republic of Korea who   were recruited during January of 2017. Data were collected via distribution   of self-report questionnaires to all nurses in the hospital who expressed   interest in participating in the study. Of the 110 surveys distributed, a   total of 99 surveys were returned. All 99 surveys were usable for further   data analysis without problems concerning missing data nor multiple   responses. Of the 99 respondents, the mean age was 30 years (SD=7.17) and   98.0% were women. Majority of participants were staff nurses (n=90, 90.9%)   who had been working in the current position between 12 to 60 months (n=51,   51.5%). Their work pattern consisted mostly of triple shifts (n=89, 89.9%),   whereas nine participants (9.1%) reported a non-shifting work schedule (Table   1). The work shifts consisted of day shift (7:30-15:00 hours), evening shift   (15:00-22:00 hours), and night shift (22:00-7:30 hours).”

Table   1.   Summary of Demographic Characteristics (N=99).

Characteristics  n (%)Age ≤2965 (65.7)30-3923 (23.3)≥4011 (11.0)Education     level Associate     degree21 (21.2)Bachelor     degree63 (63.7)Attending     graduate school4 (4.0)Master or     doctoral degree10 (10.1)Other1 (1.0)Position     department General ward72 (72.7)Intensive care     unit10 (10.1)Other17 (17.2)Length     of work experience in current position (months) <1228 (28.3)12-6051 (51.5)61-12015 (15.1)≧1216 (6.1)Work     schedule Non-shift/day     shift9 (9.1)Double shift1 (1.0)Triple shift89 (89.9)Current     position Staff nurse90 (90.9)≧Charge nurse9 (9.1)

4.    You mention that   your paper should provide guidance for "developing a strategy for   enhancing nurses' psychological health”.    I would like to hear more discussion of what you envision in this   area.

We   thank the reviewer for such helpful suggestions. We have provided examples   related to “specific intervention strategy” on page 7, line 230 as follows:

“Supplementing clinically validated effective   interventions for depressive and anxiety disorders such as cognitive behavior   therapy (CBT) with such emotion regulation training components may further   effectively reduce nurses’ psychological discomfort such as anger, depressive   and anxiety symptoms.”

5.    I think a   separate section outlining the limitations of the study and providing more   discussion on your call for future directions of research would be   helpful.  Not a big issue, but I think   a bit more discussion here would help.

We thank the reviewer for   such a suggestion. We have after careful discussion, concluded that the   present limitations and suggestions for future research are currently   logically connected in writing. However, to make this more apparent, we have   separated our discussion regarding study limitations and suggestions for   future research and added more discussion as follows:

“5. Limitations and future directions

As reported in the   meta-analysis results of Hülsheger and Schewe, there may be a significant   moderating effect of cultural difference in the relationship between   emotional labor and psychological distress [28]. Eastern   cultures are characterized as an interdependent culture that values   conformity and suppression of emotional expression for the sake of group   harmony is frequently demanded [29]. Thus, there may exist a   mediating effect of cultural differences on the sequential mediating effect   of anger suppression and anger rumination found in the present study. Hence,   the current study’s results need to be replicated through cross-cultural   research. If a difference is found, research identifying the cultural factor   that contributes to the difference also needs to be conducted.

There is a   possibility that the university hospital worked as a protective factor   compared to other work environments. According to previous studies, work   organization, structure, and policy have a significant impact on employees’   anger [30].   University hospitals have relatively well-established organizational support   and protective system that may have worked in some degree as a protective   factor for the nurses’ anger [10]. Thus, examining organizational   characteristics of nurses working at private hospitals and other emotional   laborers that may influence the result is suggested for further research.

Also, since the current study was conducted using   cross-sectional data, we cannot confirm long-term variation trends in anger   suppression and rumination and subsequent depression and anxiety.   It   is not for certain whether it is a temporary or long-term effect that anger   suppression and rumination have on depressive and anxiety symptoms, as Takebe   et al. longitudinally examined the relationship between anger suppression and   anger rumination [9]. Thus, future studies that also examine their long-term   effects on depressive and anxiety symptoms are suggested, based on our finding   that these variables are significantly associated.”

Round 2

Reviewer 1 Report

Thank you for your improvements. Regarding reference 11, please, provide in English as an appendix a summary of the article and if possible also the questionnaire.

Author Response

Dear Reviewer #1,

In response to your inquiry regarding Reference #11 ("Regarding reference 11, please provide in English as an appendix a summary of the article and if possible also the questionnaire."), we believe that the reviewer is referring to Reference #10 (Chon, Hahn, Lee, & Spielberger, 1997). Reference #11 is the English original validation study of the State-Trait Anger Expression Inventory; STAXI which is available in English publication format. If the reviewer is referring to Reference #10 which is the Korean validation study of the STAXI, the authors concluded that it may not be customary to include an article summary as an appendix in a peer-reviewed journal. Access to Reference #10 will provide an English abstract of the study as part of the article publication in the Korean Journal of Health Psychology. In regards to the reviewer's request for including the entire questionnaire, we felt that for copyright reasons, print of the STAXI-K in its full form may not be appropriate. 

As such, we have added sample items of the STAXI-K in our Measures section on page 3 lines 105-106 as follows: "Sample state anger scale items of the STAXI-K are "I am angry," and "I yell." 

Thank you for your consideration.